# Integrating Metabolomics and Transcriptomics to Unveil Atisine Biosynthesis in *Aconitum gymnandrum* Maxim

**DOI:** 10.3390/ijms232113463

**Published:** 2022-11-03

**Authors:** Lingli Chen, Mei Tian, Baolong Jin, Biwei Yin, Tong Chen, Juan Guo, Jinfu Tang, Guanghong Cui, Luqi Huang

**Affiliations:** State Key Laboratory Breeding Base of Dao-di Herbs, National Resource Center for Chinese Materia Medica, China Academy of Chinese Medical Sciences, Beijing 100700, China

**Keywords:** *Aconitum gymnandrum*, atisine, diterpene synthase, cytochrome P450

## Abstract

Diterpene alkaloids (DAs) are characteristic compounds in *Aconitum*, which are classified into four skeletal types: C_18_, C_19_, C_20_, and bisditerpenoid alkaloids. C_20_-DAs are thought to be the precursor of the other types. Their biosynthetic pathway, however, is largely unclear. Herein, we combine metabolomics and transcriptomics to unveil the methyl jasmonate (MJ) inducible biosynthesis of DAs in the sterile seedling of *A. gymnandrum*, the only species in the *Subgenus Gymnaconitum* (Stapf) Rapaics. Target metabolomics based on root and aerial portions identified 51 C_19_-DAs and 15 C_20_-DAs, with 40 inducible compounds. The highest content of C_20_-DA atisine was selected for further network analysis. PacBio Isoform sequencing integrated with RNA sequencing not only provided the full-length transcriptome but also their response to induction, revealing 1994 genes that exhibited up-regulated expression. Further, 38 genes involved in terpenoid biosynthesis were identified, including 7 diterpene synthases. In addition to the expected function of the four diterpene synthases, AgCPS5 was identified to be a new *ent*-8,13-CPP synthase in *Aconitum* and could also combine with AgKSL1 to form the C_20_-DAs precursor *ent*-atiserene. Combined with multiple network analyses, six CYP450 and seven 2-ODD genes predicted to be involved in the biosynthesis of atisine were also identified. This study not only sheds light on diterpene synthase evolution in *Aconitum* but also provides a rich dataset of full-length transcriptomes, systemic metabolomes, and gene expression profiles, setting the groundwork for further investigation of the C_20_-DAs biosynthesis pathway.

## 1. Introduction

*Aconitum* L. is a widespread genus in the Ranunculaceae family, which is further subdivided into subgen. *Aconitum*, subgen. *Paraconitum*, subgen. *Gymnaconitum*. *A. gymnandrum* maxim is the only species in subgen. *Gymnaconitum*, and is found in Tibet, western Sichuan, Qinghai, and southern Gansu provinces of China. The entire plant of *A. gymnandrum* can be used medicinally, and has essential functions as an analgesic [1], anti-inflammatory, and tumor inhibition [2], for its active substance diterpenoid alkaloids (DAs). At the same time, as a common species on the alpine meadows in the eastern Qinghai Tibet Plateau, it has been widely used as a model for studying morphological changes in arid habitats [3,4,5], including their pollination, ecology, and population genetics [6,7,8,9]. However, due to the abundant DAs in *A. gymnandrum*, it became a significant toxic plant in the Northwest Sichuan Plateau and severely hindered the grassland ecological balance and animal husbandry growth. Therefore, understanding the biosynthesis of DAs is critical to enhance *A. gymnandrum* resource utilization and sustain grassland ecological equilibrium.

DAs are the main pharmacologically active and toxic compounds in *Aconitum*, with more than 1500 compounds identified [10,11,12]. DAs are classified into four skeletal types: C_18_, C_19_, C_20_, and bis-diterpenoid alkaloids. The complexed C_18_-DAs and C_19_-DAs were proposed to be produced by the rearrangement of C_20_-DAs [11,13,14]. Therefore, an analysis of the biosynthetic pathway of C_20_-DAs is the basis for studying the structural modification of complex DAs. Like other diterpenoids, DAs were thought to be produced from the diterpene precursor geranylgeranyl pyrophosphate (GGPP), which is synthesized through IPP and DMAPP produced from the mevalonate (MVA) pathway in the cytoplasm and the methylerythritol 4-phosphate (MEP) pathway in the plastid. Copalyl diphosphate synthase (CPS) and kaurene synthase-like (KSL) enzymes then serially catalyze GGPP to form *ent*-copalyl diphosphate (*ent*-CPP) and *ent*-atiserene or *ent*-kaurene, which establishes the skeleton structure of atisine and napelline type C_20_-DAs in *A. carmichaelii* [15]. Atisine-type alkaloids are considered to be the simplest group of C_20_-DAs [10]. Atisine was found to be widely distributed in *Aconitum* species [13,16], and it was also the hotspot component for the total synthesis of C_20_-DAs [17,18].

As a marker compound in *A. heterophyllum*, atisine’s complete biosynthesis was proposed by connecting glycolysis, MVA/MEP, serine biosynthesis, and diterpene pathways [19]. As for the main diterpene pathway, it was predicted that *ent*-CPP was the only precursor, and hence two distinct biosynthetic pathways were necessary to form atisine. One was the formation of *ent*-kaurenol, *ent*-kaurenal, and *ent*-kaurenoic acid from *ent*-kaurene under continuous oxidation catalyzed by kaurene oxidase, followed by hydroxylation of *ent*-kaurenoic acid by kaurene hydroxylase to form steviol. The other pathway was to form atisenol from *ent*-atiserene by unpredicted enzymes. Finally, steviol and atisenol reacted with ethanolamine from the decarboxylation of serine to form the end product atisine (Appendix A). The study provided a panorama of atisine biosynthesis, but many details remained unclear, such as how many diterpene synthases were involved and what type of enzyme was responsible for the key hydroxylation of atisine.

Besides *A. heterophyllum*, we found other species that also had the ability to synthesize high levels of atisine [16]. By comparing the chemical constituents in seven *Aconitum* species, we found the relative content of atisine in *A. gymnandrum* and *A. tanguticum* was more than 11-fold higher than in *A. carmichaeli*, *A. vilmorinianum*, *A. stylosum*, *A. sinomontanum*, and *A. pendulum*. The level of atisine in a cell line derived from the root was 264.82 μg/g, and the content in the root of sterile seedlings was as high as 595.79 μg/g [20]. *A. gymnandrum* showed a high ability to synthesize atisine in both plants and cell lines. Thus, *A. gymnandrum* is another ideal plant to study C_20_-DAs biosynthesis pathways, with the exception of *A. heterophyllum* [19]. In the current study, methyl jasmonate (MJ) was used to induce the sterile seedling of *A. gymnandrum*. We combined transcriptome and metabolome data to explore the essential enzymes involved in the biosynthesis pathway of atisine. More importantly, we have functionally identified five diterpene synthases in *A. gymnandrum* and found a new *ent*-8,13-CPP synthase gene in *Aconitum*. These data provide a foundation for further analyzing the biosynthetic pathway of diterpenoid alkaloids.

## 2. Results

### 2.1. Metabonomic Analysis of Sterilized Seedlings in A. gymnandrum

Ultra-high performance liquid chromatography quadrupole time of flight mass spectrometry (UPLC-Q-TOF-MS) was performed to evaluate the effect of the metabolites induced. After data processing, a total of 6493 MS fragments data were obtained. The peak areas of the primary metabolites were uniformly corrected by the internal standard berberine (*m/z* 336.1234, Rt 7.63 min), and the mass spectrometry data with relative content lower than 0.001 in all samples were filtered out. Then, Masslynx 4.1 (Waters, Milford, MA, USA) was used to obtain the MS/MS fragment ions, and 77 main compounds were obtained via the merge ion fragments function for ions with the same retention time from QI (Waters, Milford, MA, USA).

According to the MS fragmentation pattern of diterpene alkaloids, which we have established in the seven *Aconitum* species [16], the compounds in seedlings were divided into three groups, 51 C19-DAs, 15 C20-DAs, and 11 non-DAs (Appendix A), of which seven of them were detected in the wild plant [16]. The most common DAs, such as aconitine, hypaconitine, mesaconitine, talatisamine, and condelphine [21,22] were not detected in the seedlings. Thus, the lack of reference substances resulted in inaccurate identification of most compounds. Only atisine (*m/z* 344.2596, Rt 3.59 min) was identified through comparison to the reference substance (Appendix A).

Although C_19_-DAs were the most abundant compounds in *A. gymnandrumn*, their relative content was not high. Besides compound **45** (*m/z* 558.3089, Rt 10.48 min) and compound **28** (*m/z* 506.3127, Rt 6.76 min), which were 7.5 and 5.0 before induction, respectively, the remaining C_19_-DAs were all lower than 2.0. On the other hand, the relative content of atisine was higher in both root and aerial portions, and the average relative content before induction was 20.1 and 10.8, respectively. Two other predicted C20-DAs, compound **56** (*m/z* 300.2331, Rt 14.24) and compound **66** (*m/z* 314.2484, Rt 17.67), also had high relative content in the root, with 24.2 and 14.1 before induction, respectively (Appendix A). These results indicate that the seedlings possess robust activity in the biosynthesis of C20-DAs, and are a suitable material for further analyzing their biosynthetic pathways.

### 2.2. Induction Effect of MJ on Metabolites in A. gymnandrum

The correlation analysis of all samples showed that the R^2^ values of root and aerial repeated samples at each induction time were almost greater than 0.8 (Appendix A). PCA analysis indicated that the root and aerial portions could be clearly separated before induction, and the induction effect of the root at different times was more obvious (Figure 1a). Five metabolites made the most significant contribution to the first major component in root. Four metabolites were C_20_-DAs, including atisine, compound **56** (*m/z* 300.2331, Rt 14.24 min), compound **63** (*m/z* 314.2483, Rt17.4 min), and compound **66** (*m/z* 314.2484, Rt 17.67 min) (Figure 1b). The only C_19_-DA observed was compound **45** (*m/z* 558.3089, Rt 10.48 min). All four C20-DAs increased more than 2-fold after induction, while the C19-DA compound **45** was not obviously accumulated after MJ induction.

MJ significantly increased the relative content of 12 C_20_-DAs, accounting for 80% of total C_20_-DAs. The initial concentration of all C_20_-DAs was higher than 0.1, two days after induction, the relative content of 12 C_20_-DAs increased 2-fold, and compound **70** (*m/z* 318.0977, Rt 18.18 min) had the highest inducible effect (14.4-fold at day 2) in C_20_-DAs. Additionally, a total of 28 C_19_-DAs were also induced at different times in the root (Figure 2c), accounting for 54.9% of the total C_19_-DAs. Compound **28** (*m/z* 506.3127, Rt 6.76 min) and compound **45** (*m/z* 558.3089, Rt 10.48 min) were the main representative compounds. The initial content of these two compounds in the roots was very high, 7.54 and 5.00, respectively, but induction only increased this content by less than 1.5-fold. Compound **35** (*m/z* 532.2929, R_t_ 8.27 min) and compound **62** (*m/z* 614.3697, Rt 17.14 min) had the highest inducible effect, which reached 0.67 and 0.12, respectively, by day 9 from a very low content (lower than 0.01). The heat map (Figure 1c) showed that atisine with the other three C_20_-DAs gather into one branch and began to accumulate on day 2. These results indicated that C_20_-DAs, especially atisine accumulated earlier than most C_19_-DAs compounds. Atisine also had the highest relative content among all metabolites, especially on day 6 after induction (Figure 1d).

### 2.3. Transcriptomic Analysis of MJ-Induced A. gymnandrum

To explore the molecular mechanism leading to the rapid accumulation of C_20_-DAs, single-molecule real-time (SMRT) sequencing PacBio Sequel and Illumina sequencing were performed to analyze the effect of induction on the transcriptome. The root and aerial portions (non-induced, 6 h, 12 h, 24 h, and 48 h post-induction) were used to perform cDNA library construction. After quality control of the original sequencing data, a total of 83,543,966 subreads were produced with an N50 of 1509 bp and GC content of 41.77% after filtering. Then, 1,307,754 circular consensus sequencings (CCS) were obtained after the SMRT Link v8.0 pipeline processed raw sequencing data. Isoseq3 software was used to obtain full-length non-chimeric consistent transcripts, and a total of 1,274,498 full-length consensus transcripts, including 111,975 polished high-quality (HQ) and 530 low-quality (LQ) transcripts, were generated. Furthermore, 24,537 transcripts could be annotated using public databases (NR, NT, GO, COG, Swiss Prot, and KEGG) (Appendix A).

The gene expression levels of each gene were obtained by calculating the Fragments Per Kilobase of transcript sequence per Millions base pairs sequenced (FPKM) values and the genes with |log2Ratio| ≥ 1 and q < 0.05 were selected as significant differentially expressed genes (DEGs). Before induction, there were 1254 genes were identified as up-regulated genes in root compared with aerial portions. After induction, a total of 601, 496, 258, and 231 genes at the 6 h, 12 h, 24 h, and 48 h time points were identified as up-regulated genes in root compared to their level in the uninduced control (0 h), respectively (Figure 2). In contrast, the number of up-regulated genes in aerial portions at each time point was significantly less (Appendix A). This was consistent with the non-obvious induction of compounds in aerial portions. After deleting the duplicated genes, a total of 1994 up-regulated DEGs were obtained and 22 common genes were up-regulated in all of the above groups (Figure 2). A total of 79 DEGs were annotated as CYP450s (cytochrome P450), 34 as methyltransferases, 29 as glycosyltransferases, 21 as BAHD-type acyltransferases, 8 as *ent*-copalyl diphosphate synthases, 2 as KOs (*ent*-kaurene oxidase), 2 as BEBTs (benzyl alcohol benzoyl transferase), and one as 2-OGD (2-oxoglutarate (2OG) and Fe (II)-dependent oxygenase superfamily protein) (Appendix A). These genes are potentially involved in the formation of structural modification of the parent nucleus of DAs.

### 2.4. Identification of Full-Length Transcripts Putatively Involved in Das Biosynthesis

Structurally, the parent nuclei of atisine are derived from *ent*-atiserene. Therefore, the genes involved in the *ent*-atiserene biosynthetic pathway were analyzed, including those involved in the upstream MVA and MEP pathways to form the diterpene precursor GGPP and the diterpene synthases to form *ent*-atiserene. A total of 38 candidate genes were identified to be involved in this process, including 9 genes encoding 6 enzymes in the MVA pathway,10 genes encoding 6 enzymes in the MEP pathway, one IDI, 3 FPPs, 5 GGPPs, 6 CPS, and 1 KSL (Figure 3). Besides these potentially involved in *ent*-atiserene biosynthesis, we also identified three monoterpenes or sesquiterpene synthase genes. It was found that both MVA and MEP pathway genes in the root were induced, but their expression patterns were different. MVA pathway genes were mostly expressed at 6 h and then decreased rapidly, while MEP pathway genes began to respond after 12 h. The expression levels of five class II diterpene synthase genes, including *AgCPS1*, *AgCPS2*, *AgCPS3*, *AgCPS5*, and *AgCPS6*, increased significantly after MJ treatment (Figure 4). However, the expression of *AgCPS4* increased only in the aerial portions after induction (Appendix A). Moreover, the expression of *AgKSL1* increased significantly in the root. After induction, MEP pathway genes in aerial portions showed a clear increasing trend compared with MVA pathway genes. When compared to the root, the expression level of aerial portions was substantially lower, with just 14 genes being highly expressed after treatment (Appendix A), consistent with the minimal change of chemical content in the aerial portion.

### 2.5. Functional Characterization of Diterpene Synthase from A. gymnandrum

To clearly define the role of diterpene synthases in *ent*-atiserene biosynthesis, we identify their biochemical functions through in vitro assays. The recombinant proteins of AgCPS1~6 (transcript 19173, transcript 18548, transcript 17999, transcript 19234, transcript 17416, and transcript 17291) and AgKSL1(transcript 16317) were expressed in *E. coli* (Appendix A). We first incubated AgCPSs with GGPP individually for comparison with the product of ZmCPS2. Three enzymes, AgCPS1, AgCPS2, and AgCPS4, yielded a single product (1) with identical retention time and mass spectrum to the product of ZmCPS2 (Figure 5), indicating that the product of these AgCPSs was CPP (geranyl pyrophosphate). They have then incubated with GGPP and AtKS synthase (specific to *ent*-CPP) to determine the configuration of CPP. All three AgCPSs produced *ent*-kaurene (3), indicating that the product configuration of those enzymes was *ent*-CPP. However, when AgCPS5 was incubated with GGPP, it produced a single unknown product (2), unlike other AgCPSs, the mass spectrum of peak2 was consistent with *ent*-8,13-CPP, the product catalyzed by PcTPS1 (*ent*-8,13-CPP synthase) from *Pogostemon cablin* (Figure 4) [23].

As for AgKSL1, we first incubated it with ZmCPS2 (*ent*-CPP synthases) and found that AgKSL1 could convert *ent*-CPP to *ent*-atiserene (4), which was identified by the known *ent*-atiserene synthase IrKSL4 from *Isodon rubescens* [24]. When combined AgKSL1with AgCPS1, AgCPS2, AgCPS4, and AgCPS5, they produced a single product, *ent*-atiserene, even though AgCPS5 is not an *ent*-CPP synthase. In conclusion, two CPP types were identified in *A. gymnandrum* and AgKSL1 was shown to be able to catalyze both of them to generate *ent*-atiserene, which ensured the efficient biosynthesis of atisine-type C_20_-DAs in *A. gymnandrum*.

### 2.6. WGCNA Analysis of Genes Associated with the C_20_-DAs Biosynthesis Pathway

Weighted gene correlation network analysis (WGCNA) is a new network modeling method designed for studying biological networks based on pairwise correlations between variables [25,26]. This method allows researchers to find the co-expressed gene modules, and explore the relationship between gene networks and phenotypes of interest, and the hub genes in the networks. We used WGCNA analysis to explore candidate genes involved in atisine biosynthesis. A total of 11,661 genes (FPKM > 10 in three biological replicates) in the root were used to perform the co-expression analysis. The MAD (median absolute deviation) of each gene was calculated using the gene expression profile and the top 4000 genes were selected to analyze. WGCNA was further used to build the scale-free co-expression network, and six co-expression modules were finally obtained, each module contained 270~1107 genes. The modules were color-coded, as shown in Figure 5a. The comparison of modules genes (Figure 5b) revealed four related module groups, which displayed similar expression profiles in different samples: (i) MEblue; (ii) MEbrown, MEgreen; (iii) Meyellow; (iv) MEturquoise, MEred (Appendix A).

The application of exogenous MJ usually activates the jasmonate (JA) signaling pathway in plants [27]. Here, three genes, transcript 72154, transcript 65043, transcript 47202, from TIFY transcription factor family, which usually acted as repressors to release targeted transcriptional factors in the JA signaling pathway were identified in module green [28]. The expression levels of these genes were significantly up-regulated to about 8-fold than the untreated control. Additionally, genes involved in MJ biosynthesis and responsive were also identified in this module, such as AOC (allene oxide cyclase, transcript 66981) and OPR (12-oxophytodienoate reductase, transcript 49184).

Deep analysis revealed that the green module contained a total of 298 genes. *AgCPS1* and *AgKSL1*, together with two *DXS*, 1 *DXR* and 1 *HDS* in MEP pathway, and 1 *GGPPS* were assigned in this module. All the above genes were up-regulated after MJ induction. DXS gene (transcript 21356) had the highest induction effect (more than 9.6-fold up-regulated at 12 h) compared to the untreated samples. Moreover, some common post-modification enzymes of DAs, including 9 CYP450s and 7 2-ODDs (2-oxoglutarate-dependent dioxygenase), were found in module green. Among them, the CYP450 gene family might play significant roles in the hydroxylation of atisine biosynthesis. A phylogenetic tree based on eight full-length CYP450s and 324 functionally identified CYP450s [29,30] showed that all of them belonged to Clan 71 (Appendix A). Seven candidate CYP450 genes (transcript 31575, transcript 28993, transcript 34646, transcript 31908, transcript 36192, transcript 36601, and transcript 38746) grouped with the CYP 71A subfamily, which were identified as potential regulators of alkaloids or flavonoids [31,32]. Transcript 34,521 and CYP706B1(from *Gossypium arboreum*) were grouped into a cluster, which involved the hydroxylation of sesquiterpene olefins [33].

### 2.7. Co-Expression Network Analysis of Module Green

To find hub genes from the module green, the function of 298 genes was further described by KEGG annotation (https://www.kegg.jp/, accessed on 8 July 2022) (Appendix A). Ninety-eight genes annotated to be involved in metabolism and environmental information processing were used to construct a co-expression network (Figure 6). It showed that the top 10 hub genes with the highest degree values included three 2-ODDs (transcript 45862, transcript 49735, transcript 49882), D-3-phosphoglycerate dehydrogenase (PGDH, transcript 19239), TIFY (transcript 65043), UDP-glucose glucosyltransferase (transcript 42527), limonene synthase (transcript 29295), citrate synthase (transcript 33670) and BAHD-type acyltransferase (transcript 49129). Two CYP450s, transcript 31,513 and transcript 31,908, were among the top 20 hub genes (Appendix A). Due to the fact that 2-ODDs were involved in the biosynthesis of various phytochemicals, including glucosinolates, flavonoids, and alkaloids [34,35]. In particular, NR (NCBI non-redundant protein sequences) described transcript 49,882 as hyoscyamine 6-dioxygenase, which catalyzed the epoxidation of scopolamine at the C6 position [36]. They may also play an important role in DA biosynthesis. PGDH was involved in the biosynthesis of serine [37], which provides the N source for atisine-type C_20_-DAs [38]. Additionally, the transcription factor TIFY (transcript 65043) showed a suitable correlation with TIFY (transcript 72154), CYP450 (transcript 36192), 2-ODD (transcript 59503), and BAHD-type acyltransferase (transcript 50810, transcript 49129). It suggested these genes could play an important role in the MJ induction process in *A. gymnandrum.*

## 3. Discussion

### 3.1. DAs from A. gymnandrum Have Different Response to MJ

MJ participates in plant biological and abiotic stress responses as a hormone and signal molecule. It is used in many plants to induce the accumulation of alkaloids, such as *D. Officinale*, *C. roseus*, and *T. wilfordii*. [39,40,41]. However, there have been no reports on the induction of diterpene alkaloids by MJ. Here, we combined metabolomics and transcriptomics to unveil the MJ-inducible biosynthesis of DAs in the sterile seedling of *A. gymnandrum*. The induction effect was evident in the root, both in the induced number of DAs and their corresponding fold-change ranges, when compared with aerial portions (Appendix A). The up-regulated genes in the root also outnumber those in the aerial portions. This may due to the fact that we mainly dripped MJ onto the solid medium so that the root contacted MJ more fully. Further investigation found more C_20_-DAs (80%) induced by MJ than C_19_-DAs (54.9%). This is consistent with the inference that C_19_-DAs originated from the further rearrangement of C_20_-DAs [13,14,15]. To obtain more C_19_-DAs with noticeable induction effects, it may be necessary to use different induction concentrations or different types of elicitors for further study.

A total of 65 DAs were detected in the sterile seedlings of *A. gymnandrum*. However, the widely distributed C_19_-DAs in *Aconitum*, such as talatizidine and talatisamine, which were found in the wild plant [16] were not detected in sterile seedlings. Karakomine, 12-*epi*-napelline, and other napline-type C20-DAs [16] were also not observed. However, wild plants accumulated a comparatively higher concentration of C_20_-type DAs. Atisine, gymnandine (denadine-type C_20_ DAs), as well as C_19_ diester-type DAs (aconigymin and 14-acetyltalatisamine) also accumulated in sterile seedlings. Therefore, we speculate that the accumulation of various forms of DAs in sterile seedlings is atisine-type C_20_ DAs > denadine-type C_20_ DAs > C_19_-DAs, implying atisine-type C_20_ DAs are the most primitive DAs in *A. gymnandrum*, which highlights its role as precursors for other C_19_-DAs.

### 3.2. New Diterpene Precursor Was Found in A. gymnandrum

Six full-length CPS genes were obtained from the full-length transcriptome. After induction of MJ, the expression of *AgCPS1*, *AgCPS2*, *AgCPS3*, *AgCPS5*, and *AgCPS6* were up-regulated both in root and aerial portions, while *AgCPS4* was only up-regulated in aerial portions (Appendix A). *AgCPS4* also had higher expression in aerial portions than in root, which suggests that AgCPS4 was responsible for the biosynthesis of DAs in the aerial portions of *A. gymnandrum*. Functional identification revealed that AgCPS1, AgCPS2, and AgCPS4 can catalyze GGPP to produce *ent*-CPP, which was previously regarded as the sole precursor for all DAs [13,14,42]. However, when we incubated AgCPS5 with GGPP, a new diterpene *ent*-8,13-CPP was identified. This *ent*-8,13-CPP synthase was only reported in *Pogostemon cablin* from *Lamiaceae* [23]. Further combination with AgKSL1 showed all four CPSs could produce *ent*-atiserene, which formed the skeleton of atisine-type C_20_ DAs. Thus, besides *ent*-CPP, we found a new precursor *ent*-8,13-CPP potentially involved in DA biosynthesis, which highlights the diverse diterpene synthase evolution in *Aconitum*. Neither AgCPS3 nor AgCPS6 could catalyze GGPP to produce any product, thus more substrates need to be tested to verify their functions.

Besides the different induced expression patterns, the diterpene synthase genes also had different expression levels in root and aerial portions (Appendix A). *AgCPS1* and *AgCPS5* were mainly expressed in the root. *AgCPS2*, *AgCPS3*, and *AgKSL1* had relatively higher expression levels in root than in aerial portions, while *AgCPS4* and *AgCPS6* were mainly expressed in aerial portions, which suggests they could have different physiological roles in plants. The highest expression level of *AgCPS1* and *AgKSL1* provides them more opportunities for the biosynthesis of DAs in the root. Compared with three different functional KSL enzymes in *A. carmichaelii* [15], we only identified AgKSL1 in *A. gymnandrum*. The fact that AgKSL1 could combine with all four identified CPS enzymes to produce the single diterpene *ent*-atiserene highlighted the possibility that *ent*-atiserene could be the main precursor in atisine biosynthesis in *A. gymnandrum*. The absence of napline-type C20 DAs in the seedling coincides with with the lack of an identified *ent*-kaurene synthase, which was regarded as the precursor of napline-type C20 DAs [14,15]. However, the *ent*-kaurene-mediated gibberellin phytohormones biosynthesis is universal in the plant kingdom [43], thus more KSL genes need to be further identified.

### 3.3. Establishing the Atisine Biosynthesis Pathway in A. gymnandrum

In comparison to the previously predicted atisine biosynthesis in *A. heterophyllum*, we found *ent*-atiserene could be the key precursor of atisine in *A. gymnandrum*, which is produced from two different intermediates, namely *ent*-CPP and *ent*-8,13-CPP. Based on these results, we outlined a detailed biosynthetic pathway of atisine in *A. gymnandrum*. First, *ent*-atiserene undergoes series-step oxidation by *ent*-kaurene oxidase-like enzymes to form intermediates *ent*-atis-16-en-19,20-diol and *ent*-atis-16-en-19,20-dial (*ent*-atisanes). L-serine is then catalyzed by a decarboxylase to form ethanolamine and further reacts with *ent*-atis-16-en-15-ol-19,20 dial to form the diterpene alkaloid intermediate 1 under the action of transaminase [38]. Then, the carbonyl group at C20 of intermediate 1 forms an N-bridge with the amino group at C19 [38], followed by dehydration and dehydrogenation to obtain intermediate 2 and dihydroatisine [44]. The hydroxylation of the C15 position is catalyzed by CYP450, but the reaction order is uncertain, and it may also be involved in catalysis after intermediate 2. Finally, atisine forms under the action of enzymes with cyclization functions such as 2-ODD (Figure 7).

In addition to the functionally identified diterpene synthases, there are five other types of enzymes including KO, 2-ODD, serine decarboxylase, CYP450, and transaminase involved in the atisine biosynthesis pathway. Among them, six CYP450 and seven 2-ODDs were screened from 194 CYP450s and 31 2-ODDs by WCGNA analysis, all fell into the green module together with AgCPS1 and AgKSL1 (Appendix A). There were three KOs, one serine decarboxylase, and two transaminases genes in the transcriptome, seven of them were induced by MJ in different degrees, which placed them in the hypothesized pathway (Figure 7). Identifying the functions of these genes by in vivo or in vitro experiments will be the goal of our next study.

## 4. Materials and Methods

### 4.1. Plant Materials and MJ Treatment

Seeds of *A. gymnandrum* were collected from Tongren County, Huangnan Tibetan Autonomous Prefecture, Qinghai, China. The seeds were inoculated on Murashige-Skoog solid medium (0.75% agar and 30 g L-1 sucrose, pH 5.8) after disinfection with 70% ethanol for 30 s and 2% sodium hypochlorite for 10 min, cultured in the dark at 25 °C until they sprouted. Then every six germinated seeds were transferred to a culture bottle (8 cm diameter, 12 cm height) containing 100 mL of the same medium. They were grown in a greenhouse for 20 days at 25 (±2 °C) under a 12 h-light/12 h-dark cycle provided by a white fluorescent lamp (1500~2000 lx). When the height of the aseptic seedling was 8–10 cm, the aseptic seedling could be used as the treatment material for MJ.

Each bottle was dripped with 2 mL MJ (0.5 mM) onto the solid medium. The sample was collected at 0 h, 6 h, 12 h, 24 h, and 48 h after MJ induction for transcriptome analysis and 0 h, 12 h, day 2, day 3, day 6, and day 9 for metabolomics analysis. Three bottles were collected for each time point. Then, the plants were divided into aerial portions (stem, leaf) and roots. Samples for transcriptome analysis were stored at −80 °C until use. The others were freeze-dried for metabolomic analysis.

### 4.2. Ditepenoid Alkaloids Extraction and UPLC-Q-TOF-MS Analysis

All the samples were dried to a constant weight under a vacuum drying process and crushed in a tissue crusher. About 10 mg of different samples were weighed, and 1.5 mL of 50% methanol (Fisher Scientific, Geel, Belgium) was added. Berberine with a concentration of 2 μg/mL was used as the internal standard. All the samples were subjected to ultrasonic treatment for 30 min, then filtered through a 0.22 μm syringe filter before analysis. The atisine reference substance provided by Li Chun from the Chinese Academy of Chinese traditional medicine was dissolved in 50% methanol.

The Acquity UPLC^TM^ system (Waters, Corp., Milford, MA, USA) was performed for Metabolite profiling. The mobile phase composition used for UPLC-Q-TOF-MS comprised a mixture of miliQ water (Millipore, Billerica, MA, USA) with 0.1% (*v*/*v*) formic acid (A) and acetonitrile (B) by an Acquity UPLC CSH C18 column (100 mm × 2.1 mm, 1.8 μm, Waters, Milford, MA, USA). The gradient of mobile phase is as follows: 0–0.1 min, 95~95%A; 0.1–3 min, 95~88%A; 3–5 min, 88~82%A; 5–8 min, 82~82%; 8–9.5 min, 82~78%; 9.5–15.5 min, 78~72%; 15.5–16 min, 72~70%; 16–17 min, 70~50%; 17–18 min, 50~20%; 18–20 min, 20~2%; 20–25 min, 2~2%; 25–25.10 min, 2~95%; 25.10–28 min, 95~95%. The injection volume was 1 μL, and the flow rate was set at 0.4 min/L. The column temperature was set at 45 °C.

The Q-TOF MS instrument used was a Synapt MS system (Waters, Corp., Milford, MA, USA). The data acquisition mode was TOF MS^E^ in positive ESI mode. The parameters were set as follows: scanning ranges from 50 to 1200 mDa. The scanning time was 0.15 s, the low-energy collision voltage was 6 V, and the high-energy collision voltage was 50–70 v. The cone voltage was 40 V, dry gas(N_2_) flow rate was 6 L/min. Data were analyzed by Masslynx 4.1 (Waters, Milford, MA, USA) and Progenesis QI software (Waters, Milford, MA, USA).

### 4.3. RNA Extraction, RNA Sequencing, and Iso-Seq Library Construction

Total RNA was extracted from different induction times in aerial portions and roots using a HuaYueYang RNA isolation kit (biotechnology, Beijing, China), and three biological replicates per sample were used. RNA samples were subjected to sequencing by a service provider (Anoroad, https://www.annoroad.com, accessed on 1 April 2020). The RNA integrity was detected using 1.0% agarose gel electrophoresis and Nanodrop (NanoDrop Technologies, Wilmington, DE, USA) was used for RNA degradation, contamination, and RNA purity. Agilent 2100 bioanalyzer and Qubit were further used to evaluate the quantified concentration of total RNA.

For RNA-seq, the samples of different time points (0 h, 6 h, 12 h, 24 h, and 48 h) after MJ treatment were performed to obtain the abundance of gene expression in different parts of the treatment time using Illumina HiseqTM2500 platform. The RNA-seq library was prepared using the TruSeq RNA Sample Prep Kit for Illumina, starting with 1 μg of total RNA. The library was purified on Beckman AMPure XP beads. The barcoded RNA-seq library was assessed by qRT-PCR using the Library Quantification Kit. The size range of the final cDNA libraries was determined on an Agilent bioanalyzer DNA7500 DNA chip (Agilent Technologies, Santa Clara, CA, USA). The cDNA libraries were sequenced on one lane for 151 cycles from each end of the cDNA fragments on a HiSeq2500 using a TruSeq SBS sequencing kit v3-HS (Illumina). The sequence images were transformed to bcl files with the Real-Time Analysis 1.17.21.2 Illumina software, which were multiplexed to fastq files with CASAVA version 1.8.2. The quality-scores line in fastq files processed with Casava1.8.2 uses an ASCII offset of 33 for presentation in the Sanger format.

SMRT sequencing was conducted using a Pacbio Sequel platform. The first-strand cDNA was synthesized using the SMARTer PCR cDNA Synthesis Kit (Clontech, Shiga, Japan), and the reverse-transcribed cDNA was PCR amplified with KAPA HiFi PCR Kits. Amplified cDNA was fractionated into 1~2 kb, 2~3 kb, and >3 kb fractions by BluePippin Size Selection (Sage Science, https://www.sagescience.com, accessed on 1 March 2020) and the SMRTbell Template Prep Kit 1.0. Libraries were used to construct three libraries of different insert sizes. Library preparation and sequencing were conducted by Anoroad (Beijing, China).

### 4.4. Iso-Seq Data Processing and Annotation

SMRTlink was used to analyze the Iso-Seq data to obtain the subreads sequence, from which the circular consensus sequencing (CCS) was extracted. Then it was divided into full-length read and non-full-length read according to the integrity of the sequence via Isoseq3. The isoform was clustered by a hierarchical n*log(n) algorithm and polished by arrow software. Finally, a high-quality consensus was obtained. The next-generation data were used to correct the consensus sequence based on LoRDEC software, and the redundancy was removed based on CD-HIT. Function annotation of identified transcripts or proteins was performed using Trinotate Release v3.2.0 (https://github.com/Trinotate/Trinotate, accessed on 20 March 2020) with default parameters, which are based on blastx, blastp, and hmmscan sequence homology searching for SWISS-PROT (A manually annotated and reviewed protein sequence database), PFAM (protein family, protein domain), KOG/COG (Clusters of Orthologous Groups of proteins), NR (NCBI non-redundant protein sequences), NT(NCBI non-redundant nucleotide sequences), GO (Gene Ontology), KO (KEGG Ortholog database) databases.

The terpenoid biosynthesis-related genes were further manually checked by aligning them with the reported genes from other species, such as *Salvia miltiorrhiza* and *Scutellaria barbata*. Besides the above annotation, the diterpene synthase genes from *A. carmichaelii* (AcCPS1, MW478118; AcCPS2-1, MW478119; AcKSL2-1, MW478123; AcKSL3-1, MW478125) were also used as the query sequence to blast the transcriptome of *A. gym-nandrum.*

### 4.5. Cloning of the Full-Length DiTPS Genes

Due to the high atisine content in the callus from the root [20], the total RNA of it was further used to verify the full length of diterpene synthase genes obtained from transcriptome assembly. One to 5 µg of total RNA was reverse transcribed into cDNA using the PrimerScriptTM RT reagent kit with a gDNA eraser (TaKaRa Corp., Dalian, China), according to the manufacturer’s instructions. pEASY^®^-Uni Seamless Cloning and Assembly Kit (TransGen Biotech, Beijing, China) was used for directly cloning the full-length cDNA into pET32 plasmid (Merck, Kenilworth, NJ, USA) for expression in Escherichia coli. The gene-specific oligonucleotides are shown in Appendix A. The plasmid was further verified by sequencing.

### 4.6. Heterologous Expression in E. coli

The plasmids with the AgCPS and AgKSL encoding gene and pET32a were transformed in prokaryotic expression strain transetta (DE3). A single positive colony was used to inoculate 200 mL of LB culture medium with 100 μg.mL^−1^ ampicillin and grown at 37 °C with shaking for OD_600_ to 0.6–0.8. Then, 40 μL 1M Isopropyl *β-D*-thiogalactopyranoside (IPTG) was added to the culture medium to induce protein expression with shaking for 14 h at 16 °C. Cells were harvested by centrifugation and resuspended in 10 mL of pre-chilled binding buffer (20 mM phosphate buffer, 137 mM NaCl, 100 mM KCl, 10 mM MgCl_2_, 2 mM DTT, 10% glycerol, pH 7.4). Cells were lysed with a regime of work for 3 s with an interval of 7 s pause to allow for cooling, a total of 15 min at 30% amplitude. Lysates were subjected to centrifugation at 12,000× *g* at 4 °C for 20 min, and soluble protein in the supernatant was mixed with the HisTrap HP Purification column. A total of 10 mL of washing buffer (binding buffer with 50 mM imidazole) and 3 mL elution buffer (binding buffer with 500 mM imidazole) were used to wash bacteria proteins and recombinant proteins, respectively.

### 4.7. In Vitro Enzyme Assay

Each recombinant AgCPS (300 μL) reacted with 5 μL of GGPP (Sigma) for 3 h at 30 °C. After the reaction, fully mixed with 1.5μL CIAP overnight in 37 °C incubators for dephosphorylating of CPP. The recombinant KSLs (400 μL) further reacted with ZmCPS2 or AgCPSs (200 μL), together with GGPP (5 μL) as substrate under the same conditions. Assay mixtures were extracted twice with 700 μL hexane. The combined hexane fractions were dried under nitrogen and resuspended in 120 μL hexane for GC-MS analysis. Four known enzymes, ZmCPS2 (Genbank: NM_001111787), IrKSL4 (KX580633), AtKS (Q9SAK2), and PcTPS1 (MH626632), were used as controls.

### 4.8. Terpene Product Analysis by GC–MS Chromatography

The terpene product was measured on a Thermo TRACE 1310 gas chromatograph with a TSQ8000 mass detector (Thermo Fisher Scientific, Waltham, MA, USA) in electron ionization mode. A capillary column TR-5ms with a 1.0 mL/min Helium flow rate (30 mm × 0.25 mm ID; DF = 0.25 µm; Thermo Fisher Scientific) using splitless injection. The GC oven temperature ramp is as follows: 50 °C, 2 min, 50 °C to 210 °C with 40 °C/min; 210 °C–250 °C with 5 °C/min; 250 °C–300 °C with 40 °C/min, with a 5 min hold at 300 °C. The ion trap temperature was 280 °C. Data analysis was performed with the device-specific software Xcalibur (Thermo Scientific).

### 4.9. Data Analyzing Software

PCA analysis was performed by Metware Cloud, which is a free online platform (https://cloud.metware.cn, accessed on 11 August 2022). Venn diagrams and Venn networks were performed using EVeen (http://www.ehbio.com/Esx, accessed on 4 July 2022) [45]. WGCNA analysis was performed using ImageGP [46]. TBtools v1.098745 [47] was used for all heatmap analyses. The Cytoscape 3.9.0 software was used to visualize the obtained network [48].

### 4.10. GenBank Accessions

The full-length transcriptome reported in this paper has been deposited in China National Center for Bioinformation under accession number PRJCA010268, which is publicly accessible for all researchers at http://bigd.big.ac.cn/gsa, accessed on 4 July 2022. GenBank accession numbers for the functional terpene synthases described in this paper are AgCPS1 (ON881911), AgCPS2 (ON881907), AgCPS4 (ON881908), AgCPS5 (ON881909), and AgKSL1 (ON881910).

## 5. Conclusions

This study used metabolomic and transcriptomic data to investigate the effects of exogenous MJ application on diterpenoid alkaloid biosynthesis in aerial portions and roots of sterile seedlings of *A. gymnandrum*. Five diterpene synthases were functionally identified, and a new diterpene *ent*-8,13-CPP was found in *Aconitum*. WGCNA was used to analyze the related co-expressed genes with atisine, and 10 hub genes were further filtered. Finally, 5 functional genes and 19 predicted genes involved in the biosynthetic pathway of atisine were established. This work predicted *ent*-8,13-CPP as a new precursor of DAs and provides a basis for further analysis of the C_20_-DAs biosynthesis pathways.

## Figures and Tables

**Figure 1 ijms-23-13463-f001:**
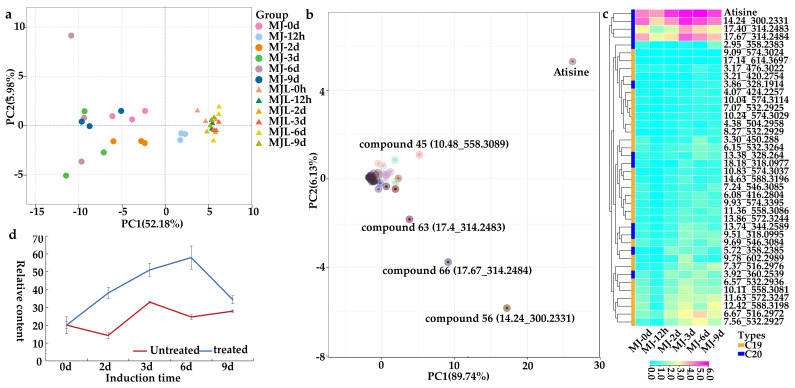
The production of Das is a major metabolic response of *A. gymnandrum* sterile seedlings to induction. MJ and MJL refer to the sample of root and aerial portions. (**a**) Principal component analysis of *A. gymnandrum of different plant portions* shows compounds with peak accumulation occurring at the indicated time, demonstrating that MJ has a greater impact on the root than aerial portions. (**b**) Principal component analysis of 77 metabolites in the root; the five compounds most representative of the first principal component are atisine, compound **45**, compound **56**, compound **63**, and compound **66**. (**c**) Cluster analysis of 40 metabolites in the root from *A. gymnandrum* at different induction times. The value of each compound was taken from relative content Log_2_ standardization. (**d**) Plots to demonstrate the increasing accumulation of the atisine in the root (error bars represent the standard error). Untreated and treated refer to the control group and experimental group, respectively.

**Figure 2 ijms-23-13463-f002:**
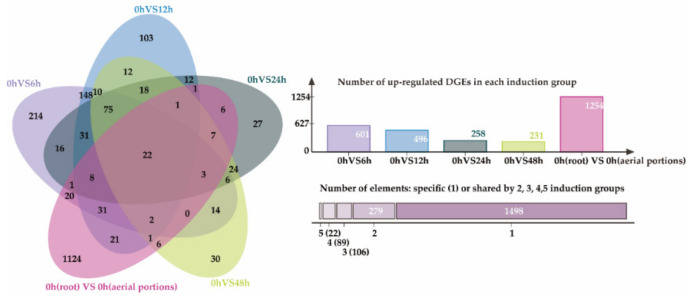
Venn diagram of up-regulated DEGs by different induction groups in the *A. gymnandrum* root.

**Figure 3 ijms-23-13463-f003:**
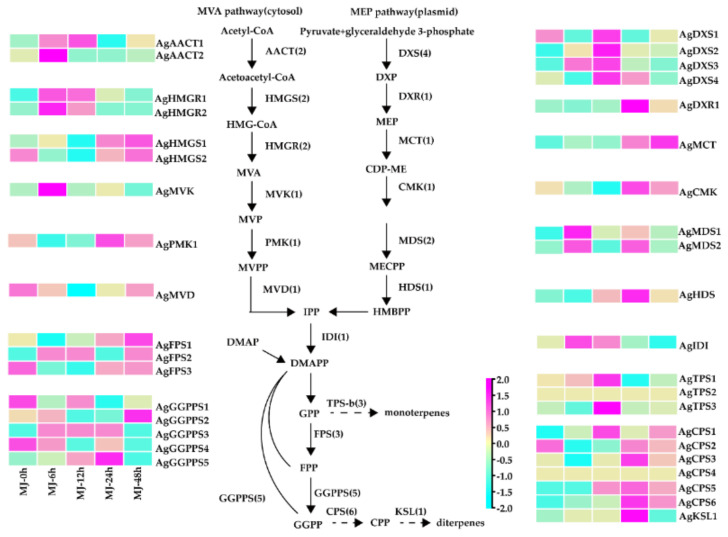
The expression profile for the genes involved in terpenoid biosynthesis in the root of *A. gymnandrum*. The transcriptome expression of each identified gene is Log2 normalization of fpkm values. Abbreviations: AACT, aceto-acetyl-CoA thiolase; CMK, 4-(cytidine 50-diphospho)-2-C-methyl-D-erythritol kinase; DXS, 1-deoxy-D-xylulose 5-phosphate synthase; DXR, 1-deoxy-D-xylulose-5-phosphate reductoisomerase; FPS, farnesyl pyrophosphate synthase; HDR, (E)-4-hydroxy-3-methylbut-2-enyl diphosphate reductase; HDS, (E)-4-hydroxy-3-methylbut-2-enyl diphosphate synthase; HMGS, 3-hydroxy-3-methylglutaryl-CoA synthase; HMGR, 3-hydroxy-3-methylglutaryl-CoA reductase; GGPPS, geranylgeranyl pyrophosphate synthase; MCT, 2-C-methyl-D-erythritol 4-phosphate cytidylyltransferase; IDI, isopentenyl diphosphate isomerase; MDS, 2-C-methyl-D-erythritol 2,4-cyclodiphosphate synthase; MDD, mevalonate diphosphate decarboxylase; MDS, 2-C-methyl-D-erythritol 2,4-cyclodiphosphate synthase; MVK, mevalonate kinase; PMK, phosphomevalonate kinase; TPS, terpene synthases (including monoterpene synthases and sesquiterpene synthases); CPS, copalyl diphosphate synthase; KSL, kaurene synthase.

**Figure 4 ijms-23-13463-f004:**
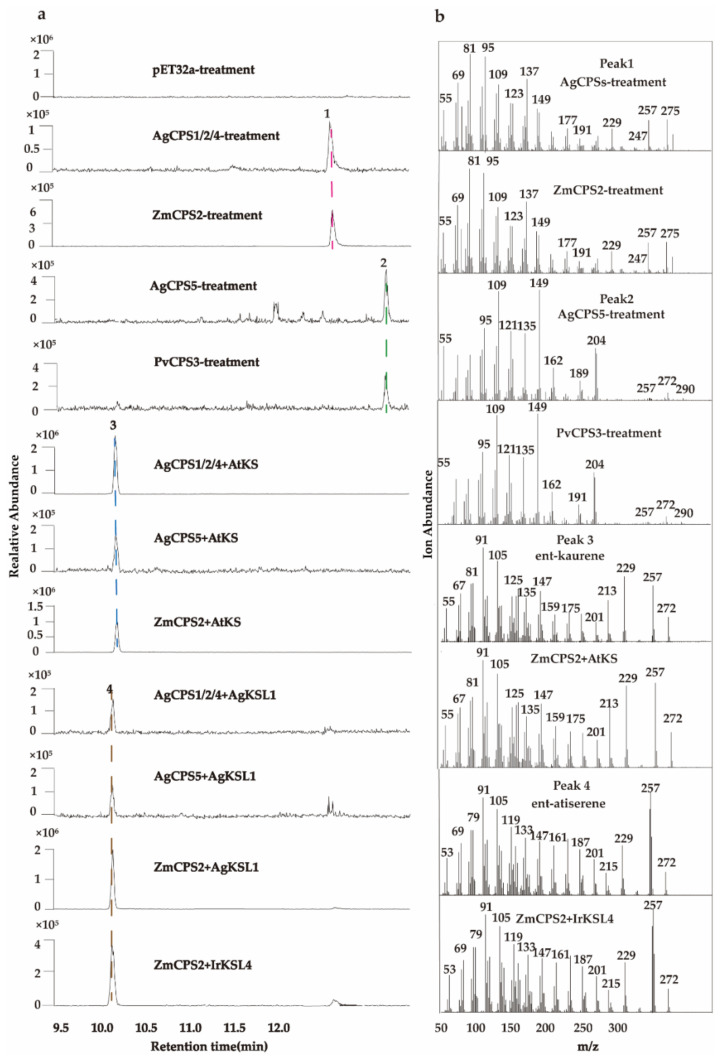
GC-MS analysis of AgCPSs and AgKSL1 reaction products obtained from in vitro assays. (**a**) 1~5: The product of AgCPS1/2/4/5 and ZmCPS2 (specific to *ent*-CPP) enzymatic reaction with GGPP, it is proved that the products of AgCPS1/2/4 and AgCPS 5 were CPP and 8,13-CPP; 6~8: The product of AgCPS1/2/4/5 and ZmCPS2 combined with AtKS (specific to *ent*-kaurene), it is proved that the products of AgCPS1/2/4 were the *ent*-configuration; 9~12: the product of AgCPS1/2/4/5 and ZmCPS2 combined with AgKSL1/IrKSL4 (specific to *ent*-atiserene), it is proved that the products of AgKSL1 was *ent*-atiserene. The enzymes ZmCPS2, PcTPS1, IrKSL4, and AtKS were used as controls. (**b**) Extracted ion chromatograms (EIC) of the product in different combinations above. Peak1: *ent*-CPP; Peak2: *ent*-8,13-CPP; Peak3: *ent*-kaurene; Peak4: *ent*-atiserene.

**Figure 5 ijms-23-13463-f005:**
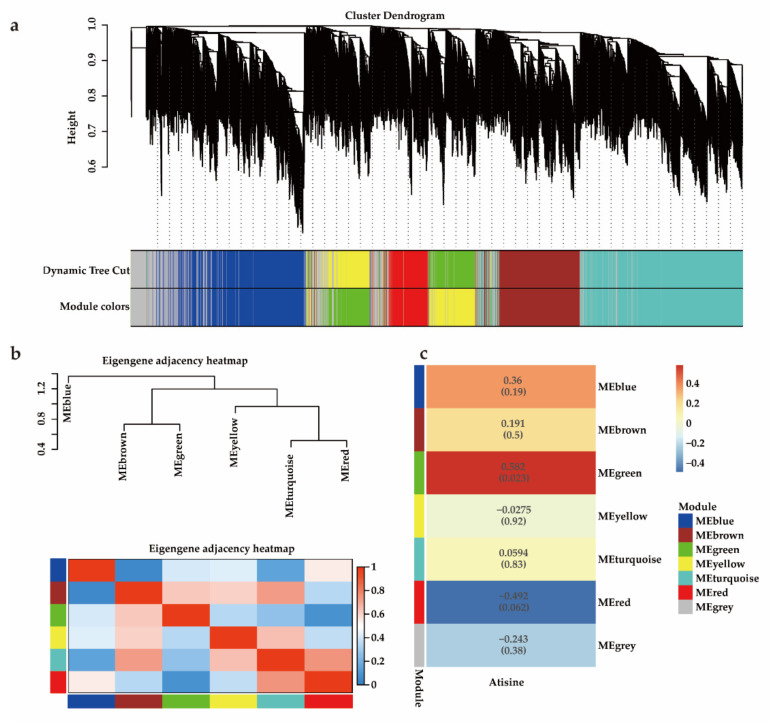
Identification of co-expression network modules in *A. gymnandrum*. (**a**) Gene dendrogram was obtained by clustering the dissimilarity based on consensus topological overlap with the corresponding module colors indicated by the color row. (**b**) Modular eigenvector clustering heatmap. The heatmap shows the relatedness of the 6 co-expression modules identified in WGCNA, with red indicating highly related and blue indicating not related. (**c**) Module-atisine correlation heatmap. The right transverse panel with red-blue indicates a color scale for module–trait correlation, from –0.5 to 0.5.

**Figure 6 ijms-23-13463-f006:**
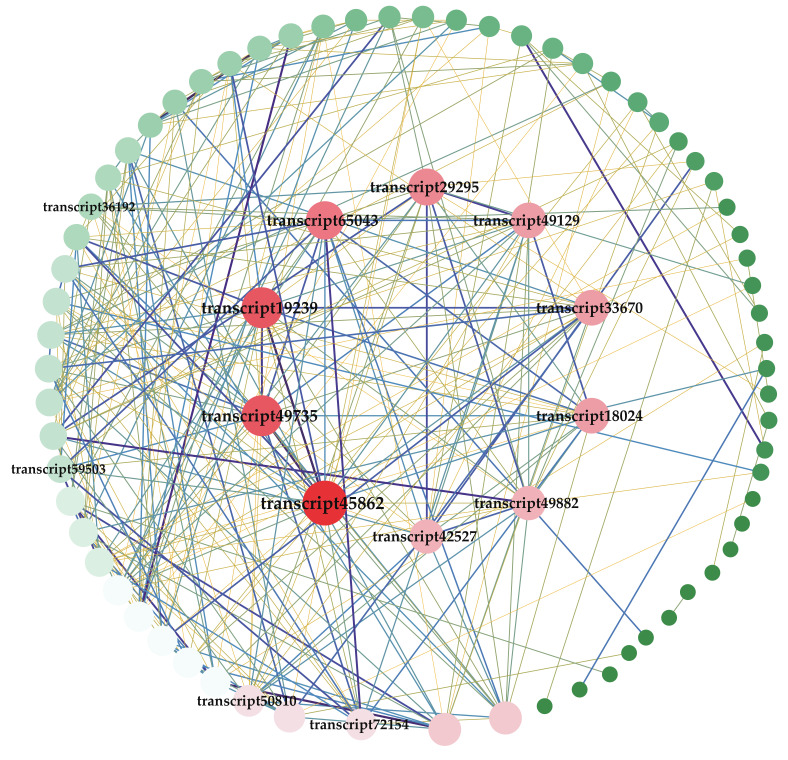
Co-expression analysis in module green of *A. gymnandrum.* Transcript 45862, transcript 49735, transcript 49882, transcript 19239, transcript 65043, transcript 42527, transcript 29295, transcript 33670, and transcript 49129 were filtered as hub genes by degree. The circle depicts genes co-expressed with 10 hub genes (r > 0.8; 77 genes in total). The size and color (red to green) of the circle were arranged according to the degree, and the lines indicate the *p*-value between the genes (purple is the largest and yellow is the smallest).

**Figure 7 ijms-23-13463-f007:**
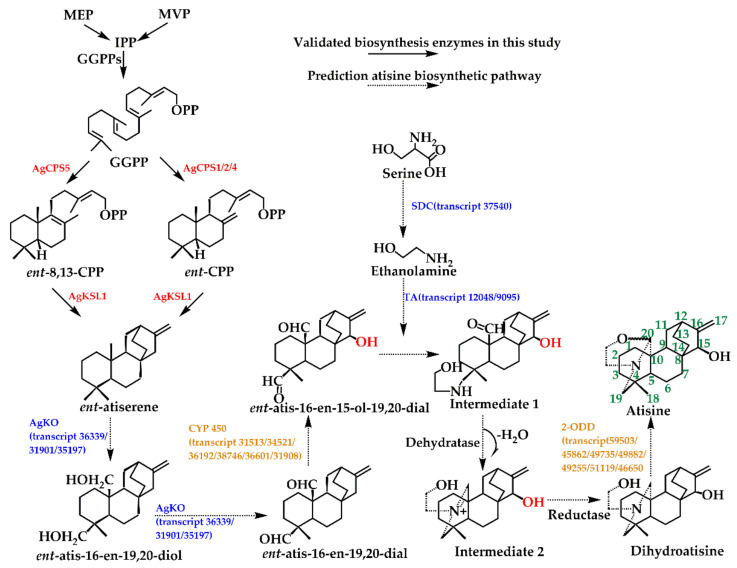
The predicted biosynthetic pathway of atisine and the involved enzyme genes in *A. gymnandrum*, red colors represent functional qualification, blue colors were screened from transcripts, and orange colors were filtered in the module green. Abbreviations: KO: *ent*-kaurene oxidase, SDC: serine decarboxylase, TA: transaminase.

## Data Availability

Transcriptome reported: http://bigd.big.ac.cn/gsa, accessed on 4 July 2022.

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
