# Peer review of "Integrating Metabolomics and Transcriptomics to Unveil Atisine Biosynthesis in Aconitum gymnandrum Maxim"

_ijms, 2022, doi:10.3390/ijms232113463_

Round 1
Reviewer 1 Report (Previous Reviewer 2)
Dear Authors!
You have significantly revised the text and added a lot of new things, which is why the scientific material presented has sparkled with the new colors. As a result, the scientific research turned out to be very interesting and filled, the various statistical calculations were especially pleasant.
With all my heart I can recommend your article for publication and I really hope that it will arose mutual scientific interest in academic community.
Best regards
Author Response
Respond: Thank you very much for your professional comments. Your recognition makes us very excited. We have learned a lot from this revision, which also gives us a new understanding of scientific research and writing. Thank you again for your support.
Reviewer 2 Report (New Reviewer)
In the manuscript 'Integrating metabolomics and transcriptomics to unveil atisine biosynthesis in Aconitum gymnandrum Maxim.' showed use transcriptomic and metabolomic data to identification genes from diterpenoid alkaloid biosynthesis pathway. The metohods were desribed suffcient, discussion and coclusions corresponding with obtained results. Few linguistic errors appear in the manuscript (eg line 93 and line 343) Question for authors is: why 11,661 genes were used to WGCNA analysis? How these genes were selected?
Author Response
Respond: Thanks for your kind advice, we have revised the linguistic errors and modified the manuscript. We obtained 24,537 genes and their FPKM values in different induction times samples through transcriptome. Then, we deleted genes with FPKM values lower than 10 in all samples and finally selected 11,661 genes for WGCNA analysis.
This manuscript is a resubmission of an earlier submission. The following is a list of the peer review reports and author responses from that submission.
Round 1
Reviewer 1 Report
Chen et al., reports the metabolic-transcriptomic analysis to uncover diterpene synthesis in A. gymnandrum. Metabolic analyses found the MeJA induction of diterpene alkaloids. Transcriptomic analyses show some genes potentially involved in the diterpene alkaloid synthesis. The research topic should be important and would attract readers who engage in secondary metabolite synthesis in plants. However, the data described are not so rigorous to support what the authors claimed, “the biosynthetic pathway of three C20-DAs and 17 related genes was established”. The experimental design of metabolite analyses are not adequate. In addition, it is critical to avoid the confusion of correlation from just transcriptomic data to causation. Obscure method description, and poorly legible data visualization must be improved.
Major comments
-Authors verified the in vitro catalytic activities of AgCPS1,2,4 as ent-CPP synthases and AgCPS5 as an ent-8,13-CPP synthase qualitatively (Fig. 5). But these results do not sufficiently support their conclusions, “Thus, besides ent-CPP, we found a new precursor ent-8,13-CPP involved in DA biosynthesis, which highlights the diverse diterpene synthase evolution in Aconitum.” in line 397 on page 13 and “This work verifies ent-8,13-CPP as a new precursor of DAs and provides a basis for further analysis of the C20-DAs biosynthesis pathways.” in line 565 in page 17. Because ent-8,13-CPP is a rarely reported metabolite in the plant kingdom as the authors mentioned in line 394 on page 13, for the statement, in addition to the expression levels of the CPS homologs (Fig. 4), the authors should include a kinetic analysis of each enzyme (Km and Vmax), knock-out (-down) analysis of each gene, or at least detection of ent-8,13-CPS from aerial parts and roots of Aconitum gymnandrum by GC-MS analysis.
-In addition, the authors used previously characterized CPS enzymes as standards instead of synthetic or commercially available pure compounds in Figure 5. However, the standard for AgCPS5 is not included in this experiment but compared to the MS fragmentation patterns in the previous report, reference21(Jhonson et al. confirmed the structure by NMR.). In MS analysis, it is often challenging to distinguish isomers. The reviewer strongly recommends the authors include PcTPS1 (reference21) as a standard for the AgCPS5 and double-check with retention time and MS fragmentation patterns.
-Methods for protein expression and in vitro enzyme assay are not described at all. Please rewrite the method section and add western blotting pictures as supplementary information.
-The methods in de novo transcriptome analysis must be improved. In addition to the more detailed annotation procedures, the annotated transcript data should be uploaded. The current description “Functional annotations of the novel genes were performed using BLAST search-520 ing against public databases such as Nr、Nt、KOG、Swiss-Prot、Pfam、KEGG、GO.” can not be reproduced.
-Overall, data visualization should be improved, especially in Figures 1 and 7. The letters on the document are too small and illegible. Network describing with too many edges only the transcript number is less informative. Please correct them otherwise remove them.
Minor comments
・In Fig.1 and others, the abbreviations such as “CK”, “CKL”, “MJL” are not defined. Please check again if all abbreviations are defined or not.
・Please change the color of all heat maps from green-red to magenta-cyan for color barrier-free.
・In all heat maps, please indicate the units of the values(such as relative intensity to the internal standard, fold Change, Log2 FC, FPKM,,,,,)
・In line 188 of page 5 and line 275 of page 9, why do the authors use both FPKM and RPKM values in this paper? Please explain the meaning.
・Figure titles of Figure2 and Supplementary Figure3 are the same. Please include the root or aerial part in the titles, respectively.
・Authors explained that they performed Methyl jasmonate induction in the manuscript, but the reviewer couldn’t catch that from the legends in figure2 and others. Please explain enough to understand what experiments were done in each Figure.
・In the legend of Fig.3, what do “different regions” mean? Please define correctly to avoid misunderstanding.
・In Figure 3 and Supplementary Figure 4, Veen->Venn.
・In all Figure legends, please use italic letters correctly, such as A. gymnandrum and else.
・In Figure4, Supplementary Figure 5, the abbreviation FPP indicates farnesyl pyrophosphate. AgFPP1,2, and 3 should be corrected to AgFPS1,2, and 3 respectively.
・In Figure 5, if you show EICs, please indicate m/z values of each chromatogram.
・In Supplementary Figure 6 and others, MVP pathway -> MVA pathway.
・Reference style is incorrect. Some are incorrect and duplicated such as reference 2 and 7, as well as 9 and 15. And even uppercase and lowercase letters are not unified such as references 21 and 22. The authors are encouraged to use reference management software like Zotero (https://www.zotero.org/).
Reviewer 2 Report
Dear Authors!
Dear Colleagues!
We read the article and found it very informative and publishing it would be very useful with perhaps some amendments.
The article makes a very favorable impression by the completeness of the review and study of the issues raised, and after minor amendments, it may well be recommended for publication in the journal.
At the same time, there are a number of questions for the authors, by answering which the authors will well complement and improve the text of the article.
1. Authors write: Atisine was found to be widely distributed in the Aconitum species [12,16]. We have previously found that atisine had the highest content in both plants [16] and cells [18] in A. gymnandrum. Another denudatine-type C20-DA gymnadine was found to be the unique component in A. gymnandrum.
We encourage authors to provide more detail about studies of atisine levels in A. gymnandrum and local advances in the research by citing their publications.
2. Authors write: C19-DAs are the main secondary metabolites of A. gymnandrum, representing more than 66.2 % of the total identified metabolites.
Where did such a statement come from? Be sure to include references to research ( articles), if available.
3. Authors write: The relative content of atisine was the highest DA in both root and aerial portions, and the average relative content before induction was 20.1 and 10.8, respectively. However, the relative content of C19-DAs was low.
How do the authors explain the achieved results, why exactly such concentrations were obtained as a result of the research? How do these figures correlate with studies by other authors on similar topics?
